# Bioavailability and provitamin A activity of neurosporaxanthin in mice

Anthony P. Miller [1], Dámaso Hornero-Méndez [2], Sepalika Bandara [3], Obdulia Parra-Rivero[4], M. Carmen Limón [4], Johannes von Lintig[3], Javier Avalos [4✉] & Jaume Amengual [1,5✉]

Various species of ascomycete fungi synthesize the carboxylic carotenoid neurosporaxanthin. The unique chemical structure of this xanthophyll reveals that: (1) Its carboxylic end and shorter length increase the polarity of neurosporaxanthin in comparison to other carotenoids, and (2) it contains an unsubstituted β-ionone ring, conferring the potential to form vitamin A. Previously, neurosporaxanthin production was optimized in *Fusarium fujikuroi*, which allowed us to characterize its antioxidant properties in in vitro assays. In this study, we assessed the bioavailability of neurosporaxanthin compared to other provitamin A carotenoids in mice and examined whether it can be cleaved by the two carotenoid-cleaving enzymes: β-carotene-oxygenase 1 (BCO1) and 2 (BCO2). Using $Bco1^{-/-}Bco2^{-/-}$ mice, we report that neurosporaxanthin displays greater bioavailability than β-carotene and β-cryptoxanthin, as evidenced by higher accumulation and decreased fecal elimination. Enzymatic assays with purified BCO1 and BCO2, together with feeding studies in wild-type, $Bco1^{-/-}$, $Bco2^{-/-}$, and $Bco1^{-/-}Bco2^{-/-}$ mice, revealed that neurosporaxanthin is a substrate for either carotenoid-cleaving enzyme. Wild-type mice fed neurosporaxanthin displayed comparable amounts of vitamin A to those fed β-carotene. Together, our study unveils neurosporaxanthin as a highly bioavailable fungal carotenoid with provitamin A activity, highlighting its potential as a novel food additive.

[1] Department of Food Science and Human Nutrition, University of Illinois at Urbana-Champaign, Urbana, IL, USA. [2] Department of Food Phytochemistry, Instituto de la Grasa, CSIC, Seville, Spain. [3] Department of Pharmacology, School of Medicine, Case Western Reserve University, Cleveland, OH, USA. [4] Department of Genetics, Faculty of Biology, University of Seville, Seville, Spain. [5] Division of Nutritional Sciences, University of Illinois at Urbana-Champaign, Urbana, IL, USA. ✉email: avalos@us.es; jaume6@illinois.edu

Carotenoids are a group of over 1000 lipophilic terpenoid compounds produced primarily by plants and algae[1,2]. The consumption of carotenoid-rich diets, as well as high plasma carotenoid levels, correlates with positive health outcomes such as a reduced incidence of cardiometabolic diseases and cancer[1,3–5]. While the bioactivities of carotenoids are diverse, those with at least one unsubstituted β-ionone ring can function as provitamin A carotenoids[6]. The most abundant provitamin A carotenoids in our diet are β-carotene, α-carotene, and β-cryptoxanthin, which together serve as the primary dietary source of vitamin A, especially for individuals with a vegetarian diet[7,8].

In mammals, carotenoids serve as substrates for either β-carotene-15,15'-dioxygenase (BCO1), β-carotene-9',10'-dioxygenase (BCO2), or both[9,10]. BCO1 is a cytosolic enzyme that catalyzes oxidative cleavage across double bonds at the 15,15' carbon position of β-carotene and is the only enzyme capable of producing vitamin A in mammals[11]. BCO2 is a mitochondrial enzyme with broad substrate specificity that catalyzes the cleavage of carotenoids at the 9',10' positions[10,12]. As wild-type mice cleave dietary carotenoids at a greater rate than humans, the utilization of $Bco1^{-/-}$ and $Bco2^{-/-}$ mice allows for the evaluation of carotenoid absorption and accumulation in rodents[12,13].

Certain heterotrophic organisms synthesize carotenoids, but it remains unexplored whether foods containing these organisms can significantly contribute to increased carotenoid levels in humans. Over the past few decades, biotechnologists have acquired the ability to harness fungi to overproduce carotenoids and other molecules by adopting optimal growth conditions such as strong sun exposure and nutrient scarcity[14]. Various *Neurospora* and *Fusarium* species produce a carboxylic, relatively hydrophilic carotenoid identified as neurosporaxanthin, which gives these fungi their characteristic orange color[15,16]. Neurosporaxanthin has spurred interest in the scientific community due to its short length and carboxylic tail, which may confer this carotenoid greater bioavailability than other carotenoids. Additionally, neurosporaxanthin contains an unsubstituted β-ionone ring, suggesting it may function as a provitamin A carotenoid in mammals.

In this study, we compared the absorption of neurosporaxanthin to β-carotene and β-cryptoxanthin in mice lacking both BCO1 and BCO2. We also assessed the ability of neurosporaxanthin to generate vitamin A using enzymatic assays. Lastly, we evaluated the contribution of daily administration of neurosporaxanthin to vitamin A production in mice and whether it resulted in toxicity. Our data show that neurosporaxanthin is absorbed in mammals at greater rates than β-carotene and β-cryptoxanthin and that this carotenoid is preferentially cleaved by BCO1 to form vitamin A.

## Results

**Production and purification of neurosporaxanthin in *F. fujikuroi* overproducing cultures.** Nitrogen scarcity is a light-independent inducing agent for carotenoid synthesis in the fungus *F. fujikuroi*, including the *carS* mutants that overproduce neurosporaxanthin[17]. Following established protocols in our lab, we grew *carS F. fujikuroi* mycelia under shaking conditions in an optimized medium with a high carbon/nitrogen ratio leading to carotenoid production by the fungus in the stationary phase under depleted nitrogen conditions[18]. After 5 weeks, mycelia produced approximately 6 mg of total carotenoids per g dry weight, of which neurosporaxanthin was the predominant component (Fig. 1a). After 500 repetitive high-performance liquid chromatography (HPLC) injections, we obtained a total of 50 mg of purified neurosporaxanthin (Fig. 1b). The identity of neurosporaxanthin was confirmed by its UV/Visible (Fig. 1c) and mass spectrometry (Fig. 1d) spectra. Based on the chromatographic data, the purity of neurosporaxanthin utilized for our experiments was over 98%.

**Neurosporaxanthin is absorbed at a greater rate than other provitamin A carotenoids.** Preclinical and clinical studies suggest intestinal carotenoid absorption and bioavailability are directly associated with their polarity[19–23]. Hence, we first compared the absorption of neurosporaxanthin, a 35-carbon xanthophyll terminated by a carboxylic acid moiety at one end and a β-ionone group at the other, with two 40-carbon carotenoids: β-carotene and β-cryptoxanthin (Fig. 2a, b inset). To rule out the influence of carotenoid cleavage on carotenoid absorption and tissue accumulation, we utilized mice lacking the two carotenoid-cleaving enzymes: BCO1 and BCO2. We primed $Bco1^{-/-}Bco2^{-/-}$ mice for carotenoid absorption with a vitamin A-deficient, carotenoid-free diet for 2 weeks before the intervention, as done in the past[24]. $Bco1^{-/-}Bco2^{-/-}$ mice were gavaged with a single dose of cottonseed oil containing 30 mg/kg body weight of either β-carotene, β-cryptoxanthin, or neurosporaxanthin. Control mice were gavaged with the same volume of vehicle (cottonseed oil). After the gavage, we collected feces every 12 h for a total of 3 days and sacrificed the mice at the end of the intervention to collect their plasma and livers (Fig. 2b).

In agreement with previous reports showing that murine intestinal transit is complete by 24 h, mice excreted the maximal amount of carotenoid 24 h post-gavage[25]. After 24 h, carotenoid levels in the feces showed a sharp decline (Fig. 2c). We estimated carotenoid absorption by quantifying the cumulative carotenoid content in feces. HPLC quantifications revealed an inverse association between carotenoid polarity and fecal concentration (Fig. 2d). We also measured plasma and hepatic carotenoid concentrations. HPLC data showed the opposite pattern observed in the feces, where neurosporaxanthin accumulated at the highest concentration, followed by β-cryptoxanthin and β-carotene (Fig. 2e, f).

**BCO1 cleaves neurosporaxanthin to generate vitamin A in mice.** We next investigated the contribution of BCO1 and BCO2 to neurosporaxanthin cleavage. To this end, we gavaged age and sex-matched wild-type, $Bco1^{-/-}$, $Bco2^{-/-}$, and $Bco1^{-/-}Bco2^{-/-}$ mice with 30 mg neurosporaxanthin/kg body weight or vehicle control for 10 consecutive days. A separate cohort of wild-type mice was dosed with 30 mg β-carotene/kg body weight. At the end of the intervention, mice were sacrificed to collect plasma and tissues.

Neurosporaxanthin did not affect food intake, body weight, liver weight-to-body weight ratio, or adiposity. Furthermore, neurosporaxanthin did not alter ALT/AST enzymatic activities or *miR-122* levels in plasma, a liver-specific miRNA only found in circulation upon liver injury[26] (Table 1).

HPLC analyses showed that wild-type and $Bco2^{-/-}$ mice accumulated approximately 25-fold lower plasma and 6-fold lower hepatic neurosporaxanthin concentrations compared to $Bco1^{-/-}Bco2^{-/-}$ mice. $Bco1^{-/-}$ mice accumulated more neurosporaxanthin than wild-type and $Bco2^{-/-}$ mice but showed 5-fold and 3-fold lower plasma and hepatic neurosporaxanthin in comparison to $Bco1^{-/-}Bco2^{-/-}$ mice, respectively. β-carotene was not detectable in the plasma of wild-type mice but appeared at a lower concentration than neurosporaxanthin in the liver of these animals (Fig. 3a, b).

We next analyzed whether the enzymatic cleavage of neurosporaxanthin results in the formation of vitamin A in mice. Circulating vitamin A levels in wild-type showed no differences between groups (Fig. 3c). We also measured vitamin A levels in

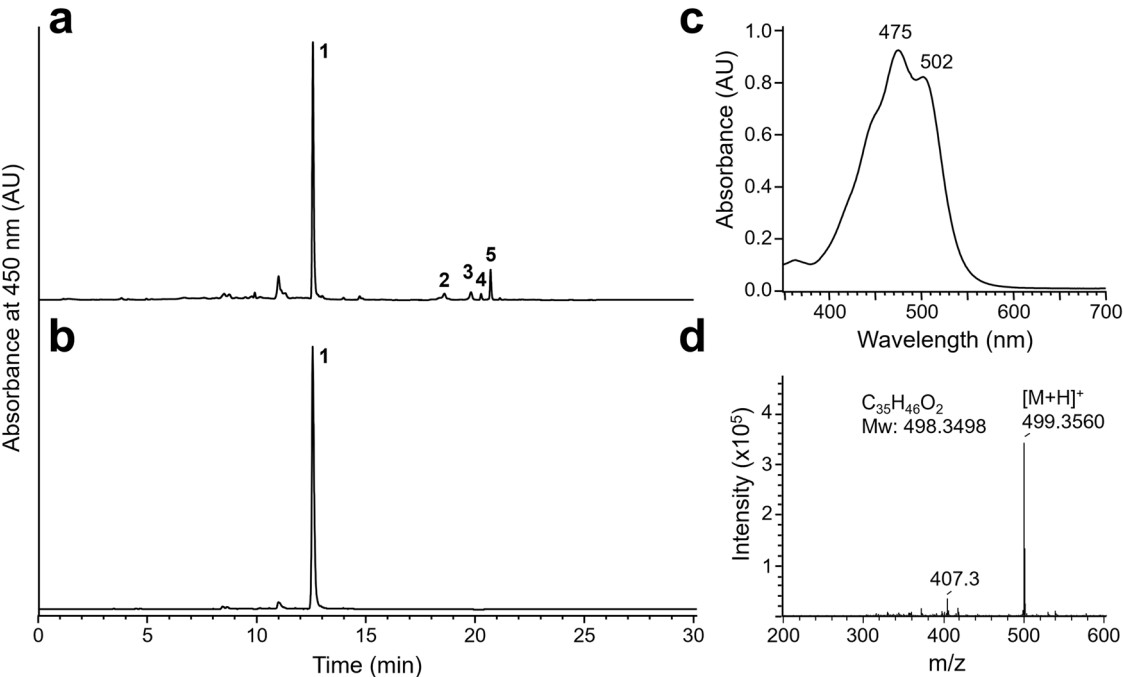

**Fig. 1 Purification of neurosporaxanthin. a** Chromatogram displaying carotenoids accumulated in the SG39 strain of *F. fujikuroi* containing neurosporaxanthin (peak 1), torulene (peak 2), γ-carotene (peak 3), ζ-carotene (peak 4), and β-carotene (peak 5) at detection wavelength set to 450 nm in a reverse-phase HPLC system. **b** Chromatogram of purified neurosporaxanthin. **c** UV/Visible spectrum of purified neurosporaxanthin showing the characteristic absorption maxima at 475 and 502 nm. **d** Mass spectrum of purified neurosporaxanthin. AU: Arbitrary units.

the liver and the WAT, two major vitamin A reservoirs in mammals[27]. Hepatic vitamin A stores remained constant between groups, but WAT retinoids showed an overall tendency towards increasing in wild-type mice fed neurosporaxanthin and β-carotene in comparison to the control group (Fig. 3d, e). Lastly, we measured retinoids in two portions of the small intestine, which represents the intestinal conversion of carotenoids to vitamin A[28]. Retinoid concentrations detected in the duodenum and jejunum of mice dosed with neurosporaxanthin or β-carotene were higher than those dosed with vehicle (Fig. 3f).

**BCO1 and BCO2 cleave neurosporaxanthin in vitro.** Our results in mice suggest that both BCO1 and BCO2 participate in the cleavage of neurosporaxanthin. To validate these results, we performed enzymatic analyses in which murine BCO1 and BCO2 fused to MBP were expressed in *E. coli*, as described previously[29,30]. The central cleavage of neurosporaxanthin by BCO1 resulted in the formation of all-*trans* retinal and 15-apocarotenal-4-carotenoic acid. Neurosporaxanthin incubation with BCO1 resulted in a drastic reduction of the xanthophyll content, as evidenced by the color shift from red to pale yellow (Fig. 4a–c).

The incubation of neurosporaxanthin with BCO2 resulted in the formation of 10'-apocarotenal-4-carotenoic acid according to the retention time and spectral characteristics of the compound. The second cleavage product, β-ionone, was not detectable because of its volatility and loss during the extraction procedure (Fig. 4d–f). Thus, recombinant BCO1 and BCO2 cleave neurosporaxanthin to form vitamin A aldehyde and non-provitamin A apocarotenoids, respectively.

**Neurosporaxanthin activates retinoic acid signaling in the small intestine in wild-type mice, but not in *Bco1⁻/⁻Bco2⁻/⁻* mice.** The structure of neurosporaxanthin, which like retinoic acid contains a β-ionone ring and a carboxylic acid moiety,

suggests it could activate gene expression via the retinoic acid receptors (RARs) (Fig. 5a). Based on our HPLC data, we selected the small intestine as the organ displaying the greatest response on vitamin A formation (Fig. 4f). In the intestine, the formation of retinoic acid transactivates RARs to upregulate the transcription factor ISX, which in turn, downregulates *Bco1* and *Sr-b1* expression[31]. Gene expression analyses revealed that neurosporaxanthin and β-carotene showed an overall trend towards the induction of *Isx* levels accompanied by a decrease in *Bco1* and *Sr-b1* expression in comparison to vehicle-treated mice (Fig. 5b). These results suggest that neurosporaxanthin intake results in the production of vitamin A in the gut, where BCO1 is highly expressed[32].

To rule out the production of vitamin A or other metabolites by the carotenoid-cleaving enzymes, we compared the expression of retinoid-responsive genes in *Bco1⁻/⁻Bco2⁻/⁻* mice gavaged for 10 days with neurosporaxanthin or cottonseed oil vehicle. Our RT-PCR results failed to detect differences in the expression levels of classical retinoic acid-responsive genes in the small intestine of *Bco1⁻/⁻Bco2⁻/⁻* mice exposed to neurosporaxanthin (Fig. 5c).

**Discussion**

Carotenoid absorption is a major limiting step in the production of vitamin A. Several factors affect carotenoid absorption, such as dietary fat content, food processing methods, and genetic variation in carotenoid-related enzymes and transporters[33]. In our first experiment, we compared the absorption of β-carotene, β-cryptoxanthin, and neurosporaxanthin in mice lacking both carotenoid-cleaving enzymes. We selected *Bco1⁻/⁻Bco2⁻/⁻* mice to rule out carotenoid cleavage in the quantification of carotenoids in feces and tissues. We observed that carotenoid polarity was directly associated with intestinal absorption (Fig. 2). This was not surprising since the physicochemical properties of carotenoids determine carotenoid absorption by modifying their orientation in the micelles formed during digestion, which affects their accessibility to enterocytes[34,35]. Relatively polar carotenoids

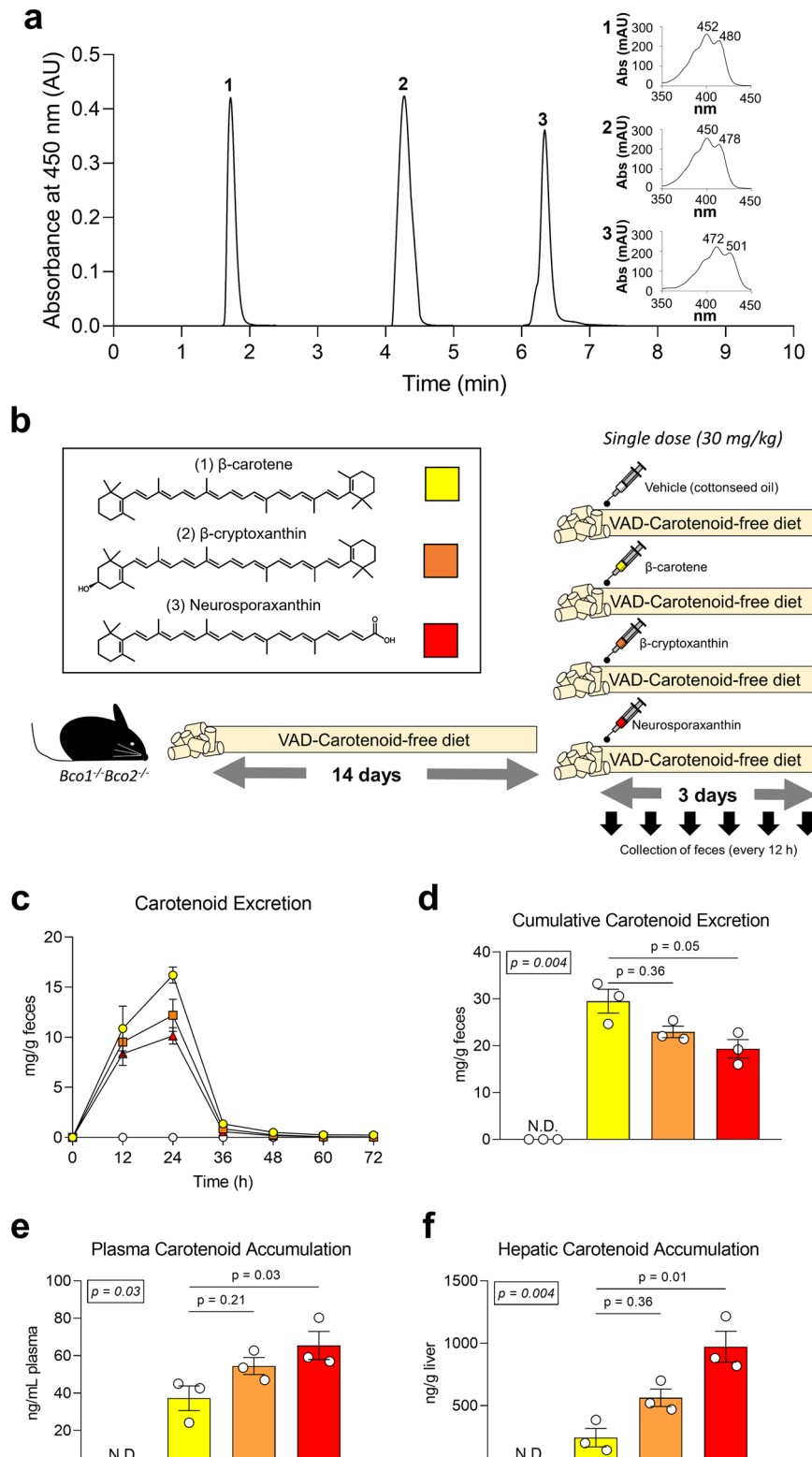

such as lutein and β-cryptoxanthin orient towards the surface of micelles and are more easily transferred to the aqueous phase, whereas non-polar carotenoids like lycopene and β-carotene are located deeper within micelles[36,37]. These findings are further supported by cell culture studies showing that putative carotenoid membrane transporters display a greater affinity for xanthophylls than for carotenes[38,39], an observation also reported in clinical settings[22,40,41]. Whether neurosporaxanthin requires SR-B1 to enter the enterocyte, as it occurs with other carotenoids[42], or traverses the plasma membrane via passive diffusion as retinoids do, is not yet known.

For all our experiments, we consistently utilized cottonseed oil as a vehicle for our gavage solutions. Cottonseed oil is frequently employed in investigations concerning carotenoids due to its minimal carotenoid content[43–45]. However, studies show that carotenoid absorption increases in oils containing

**Fig. 2 Carotenoid excretion and tissue accumulation.** Six-week-old *Bco1⁻/⁻Bco2⁻/⁻* mice were provided a purified carotenoid-free, vitamin A-free diet for 2 weeks before the administration of a single gavage of cottonseed oil vehicle, 30 mg/kg β-carotene, 30 mg/kg β-cryptoxanthin, or 30 mg/kg neurosporaxanthin. After gavage, we collected feces every 12 h for 3 days. **a** Chromatogram standards of β-carotene (peak 1), β-cryptoxanthin (peak 2), and neurosporaxanthin (peak 3) at a detection wavelength set to 450 nm separated with a normal-phase HPLC system. Insets show the corresponding spectra of peaks 1 to 3. **b** Experimental design with the structure of the carotenoids utilized in the study. **c** HPLC quantification of carotenoid concentrations in feces over 72 h post-gavage, and **d** their cumulative excretion. **e** HPLC carotenoid quantifications in plasma, and **f** liver. Values are the means ± SEMs, *n* = 3 mice/group. Statistical differences were evaluated with the Kruskal–Wallis *H*-test applying Dunn's test correction for multiple comparisons (*p* < 0.05). We display overall *p*-values (number in the box) and follow-up post hoc tests (adjusted *p*-value) in comparison to β-carotene-fed mice. AU: Arbitrary units.

**Table 1 Effects of short-term treatment with neurosporaxanthin on body weight and hepatic toxicity in wild-type, *Bco1⁻/⁻*, *Bco2⁻/⁻*, and *Bco1⁻/⁻Bco2⁻/⁻* mice[a].**

| Parameter, (units) | Vehicle | Neurosporaxanthin | Treatment *p*-value |
|---|---|---|---|
| Average food intake, (g/mouse/day) | 2.15 ± 0.05 | 2.10 ± 0.08 | 0.42 |
| Body weight, (g) | 17.8 ± 0.46 | 17.3 ± 0.48 | 0.48 |
| Liver weight/body weight | 0.04 ± 0.006 | 0.04 ± 0.006 | 0.66 |
| Adiposity[b] | 0.05 ± 0.003 | 0.04 ± 0.004 | 0.22 |
| Plasma ALT activity, (mU/mL) | 20.8 ± 1.4 | 18.2 ± 1.8 | 0.16 |
| Plasma AST activity, (mU/mL) | 28.4 ± 2.9 | 28.6 ± 2.3 | 0.96 |
| Relative plasma *miRNA-122* expression (relative to vehicle) | 1.0 ± 1.7 | 0.83 ± 1.1 | 0.77 |

*ALT* alanine transaminase, *AST* aspartate transaminase.
[a]Values are the means ± SEMs. Genotype, treatment, and interaction between both factors were evaluated by the Kruskal–Wallis *H*-test applying Dunn's test correction for multiple comparisons. We did not observe a genotype or interaction effect for any of the parameters, therefore, we displayed *p*-values exclusively for the treatment effect. mU; milliunits. *n* = 12 mice/group.
[b]Adiposity was calculated by dividing the sum of the gonadal, inguinal, and perirenal white adipose tissue weights by the total body weight.

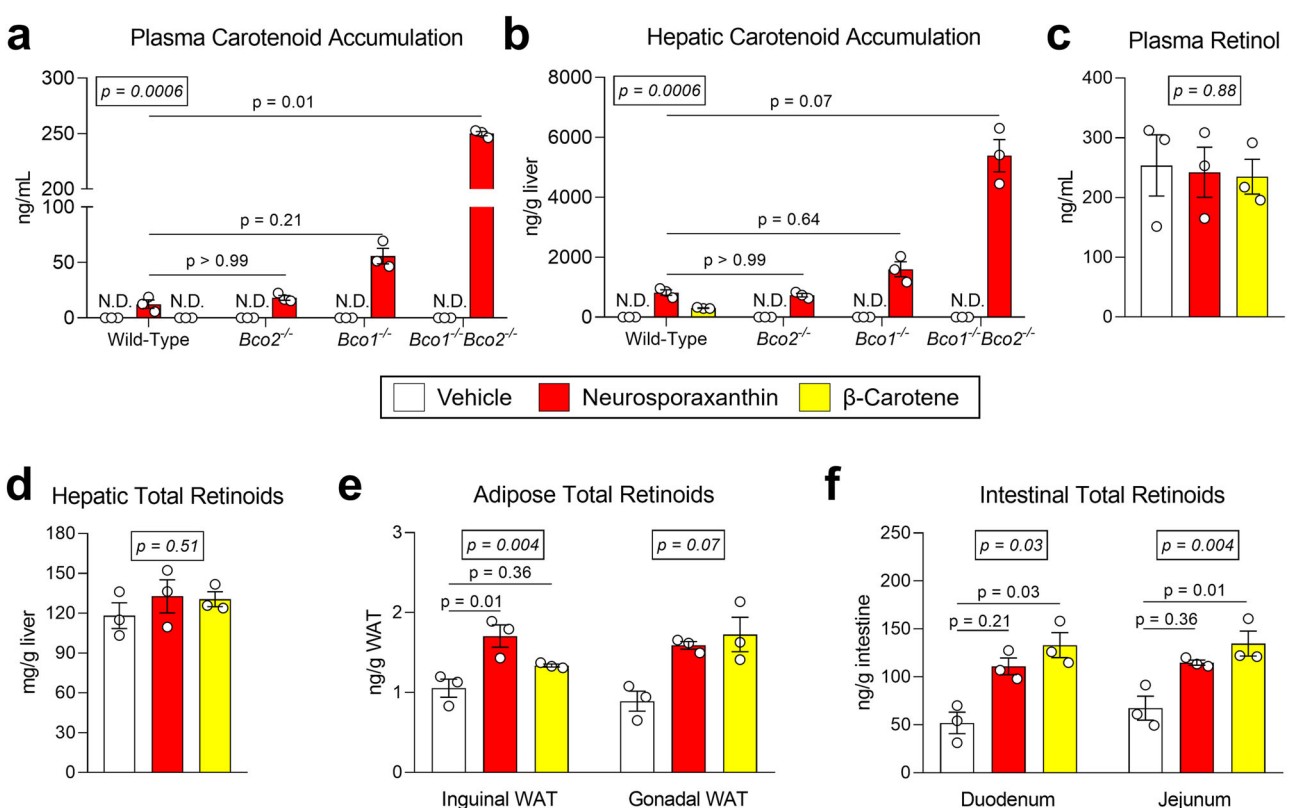

**Fig. 3 Neurosporaxanthin accumulation and vitamin A production in mice.** Six-week-old wild-type, *Bco1⁻/⁻*, *Bco2⁻/⁻*, and *Bco1⁻/⁻Bco2⁻/⁻* mice were provided a purified carotenoid-free, vitamin A-free diet for 2 weeks before the initiation of daily gavages of cottonseed oil vehicle or 30 mg/kg neurosporaxanthin for 10 days. A separate cohort of wild-type mice was gavaged with 30 mg/kg β-carotene for the same period. **a** HPLC carotenoid quantifications in plasma, and **b** liver. **c** HPLC vitamin A quantifications in plasma, **d** liver, and **e** inguinal and gonadal white adipose tissues (WAT), and **f** two portions of the small intestine in wild-type mice. Values are the means ± SEMs, *n* = 3 mice/group. Statistical differences were evaluated with the Kruskal–Wallis *H*-test applying Dunn's test correction for multiple comparisons (*p* < 0.05). We display overall *p*-values (number in the box) and follow-up post hoc tests (adjusted *p*-value) in comparison to wild-type mice fed neurosporaxanthin (**a**, **b**) or Vehicle (**c–f**).

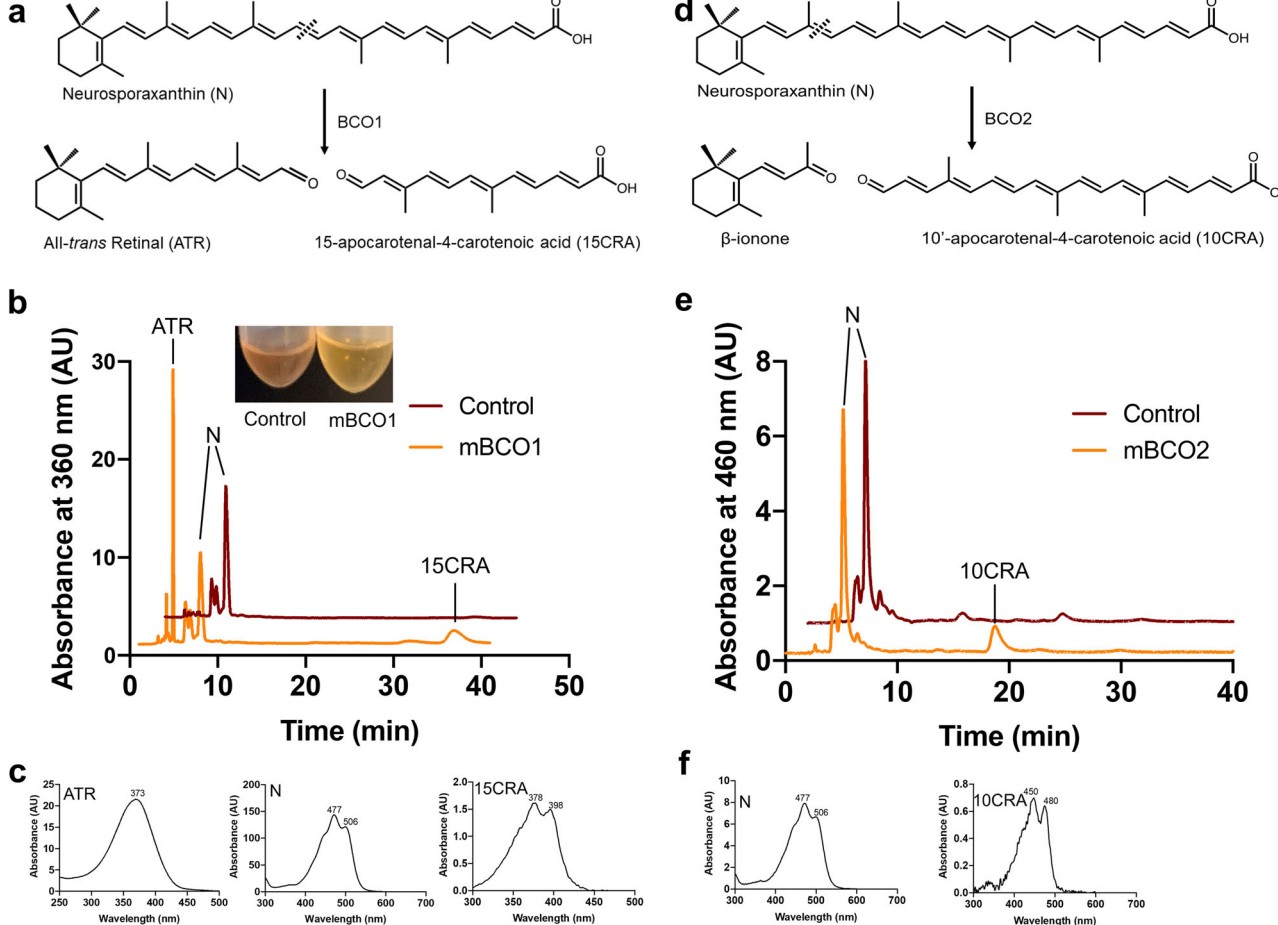

**Fig. 4 BCO1 and BCO2 cleave neurosporaxanthin at the 15,15' and 9',10' positions, respectively. a** Schematic of oxidative cleavage of neurosporaxanthin by BCO1. **b** HPLC chromatograms of lipid extracts from in vitro enzyme activity assays of murine BCO1 (orange trace) incubated with neurosporaxanthin. The red trace shows control incubations without the addition of enzyme. **c** UV/Visible spectra of all-*trans* retinal, neurosporaxanthin, and 15-apocarotenal-4-carotenoic acid. **d** Schematic of oxidative cleavage of neurosporaxanthin by BCO2. **e** HPLC chromatograms of lipid extracts from in vitro enzyme activity assays of murine BCO2 (orange trace) incubated with neurosporaxanthin. The red trace shows control incubations without the addition of enzymes. **f** UV/Visible spectra of neurosporaxanthin, and 10'-apocarotenal-4-carotenoic acid. AU: Arbitrary units, ATR: All-trans retinal, N: Neurosporaxanthin, 15CRA: 15-apocarotenal-4-carotenoic acid, 10CRA: 10'-apocarotenal-4-carotenoic acid.

monounsaturated fat such as olive oil in comparison to cottonseed oil, which is rich in polyunsaturated fats[46,47]. Whether the composition of fat affects neurosporaxanthin's uptake in the same way it alters other carotenoids remains unanswered. Regardless, we acknowledge that delivering purified carotenoids in oils probably results in greater uptake rates than when carotenoids are present in foods matrices.

This is the first study to examine the metabolism of neurosporaxanthin in mammals, prompting us to determine if BCO1 or BCO2 cleaves this acidic carotenoid and if this cleavage results in the formation of vitamin A. Studies in $Bco1^{-/-}$ mice revealed that BCO1 is required for the generation of vitamin A from β-carotene and β-cryptoxanthin[11,13]. While BCO1 displays a greater affinity for β-carotene, studies showed that purified BCO1 can also cleave β-cryptoxanthin[48]. In the case of β-cryptoxanthin, mouse studies revealed a sequential cleavage initiated by BCO2 to form β-apo-10'-carotenal, which later serves as a substrate for BCO1 to produce retinal[11]. Our findings demonstrate that BCO1 is primarily responsible for neurosporaxanthin cleavage in mice and in vitro enzymatic assays, indicating that BCO1 has a high affinity for the unsubstituted β-ionone ring in neurosporaxanthin, which is unaffected by its carboxylic acid moiety. (Figs. 3a, b and 4).

Taking into account that $Bco1^{-/-}$ and $Bco1^{-/-}Bco2^{-/-}$ mice fed β-carotene for 10 weeks accumulate the same amount of β-

carotene in tissues[11], it was surprising to us that neurosporaxanthin accumulation in $Bco1^{-/-}Bco2^{-/-}$ mice was markedly greater than in $Bco1^{-/-}$ mice (Fig. 3b, c). Hence, our data suggest that while BCO1 is primarily responsible for neurosporaxanthin cleavage, BCO2 displays a greater affinity for neurosporaxanthin in comparison to β-carotene, at least in mice. Several factors could be responsible for these differences, such as the preferential accumulation of polar carotenoids in the mitochondria, where BCO2 resides[12]. Recently, a novel class of lipid transfer proteins, the Aster proteins, were identified as intracellular carotenoid transporters[49]. The authors showed that Aster protein expression correlates with the tissue accumulation of zeaxanthin in $Bco2^{-/-}$ mice. It could be surmised that carotenoids that do not contain an unsubstituted β-ionone ring, such as lutein and zeaxanthin, are targeted for delivery to the mitochondria by Aster proteins, while those with unsubstituted β-ionone rings, such as β-carotene and neurosporaxanthin, remain accessible to cytosolic BCO1. However, comprehension of the intracellular trafficking of carotenoids is in its nascent stages, and the interplay between Aster proteins and the well-established protein constituents of carotenoid metabolism is a domain warranting subsequent inquiry.

Only a handful of carotenoids possess provitamin A activity in mammals. The presence of two β-ionone rings in the structure of β-carotene, as well as its abundance in our diet and plasma,

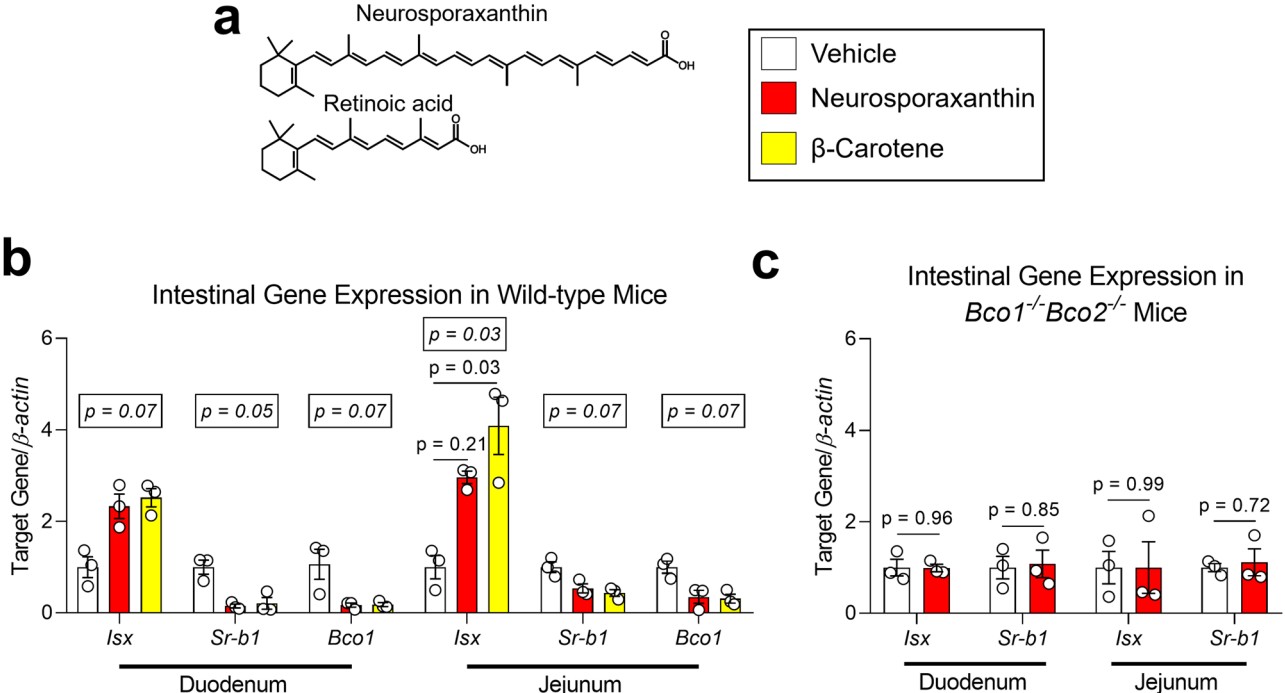

**Fig. 5 Effect of neurosporaxanthin on intestinal gene expression in mice. a** Structure of neurosporaxanthin and retinoic acid. **b** Six-week-old wild-type and $Bco1^{-/-}Bco2^{-/-}$ mice were provided a purified carotenoid-free, vitamin A-free diet for 2 weeks before the initiation of daily gavages of cottonseed oil vehicle or 30 mg/kg neurosporaxanthin for 10 days. Data show intestinal mRNA expression in wild-type, and **c** $Bco1^{-/-}Bco2^{-/-}$ mice. Values are the means ± SEMs, $n = 3$ mice/group. Statistical differences were evaluated with the Kruskal–Wallis $H$-test applying Dunn's test correction for multiple comparisons ($p < 0.05$). We display overall $p$-value (number in box) and follow-up post hoc test (adjusted $p$-value) in comparison to wild-type mice vehicle (**b**), or two-tailed Student's $t$-test (**c**).

indicates that β-carotene is the most relevant provitamin A carotenoid in the human diet[1,8]. However, studies utilizing Mongolian gerbils, a well-regarded model of human carotenoid metabolism[50], illustrate that the supplementation of β-cryptoxanthin maintains vitamin A stores equally to the supplementation of β-carotene[51,52]. Similarly, our data suggest neurosporaxanthin maintains vitamin A stores equally to β-carotene when it is provided as the only source of dietary vitamin A, highlighting the importance of carotenoid absorption in the formation of vitamin A.

The provitamin A function of neurosporaxanthin, in addition to its antioxidant properties[18], gives this xanthophyll considerable biotechnological interest as a potential food additive. While this study marks the initial instance of administering pure neurosporaxanthin to mammals, it is important to note that this carotenoid is naturally occurring in food items originating from *Neurospora* found in Eastern Asia. Specifically, examples of this occurrence are evident in soybean and okara fermentation products termed s-oncom and o-oncom, respectively, which are produced through a fermentation process involving *Neurospora intermedia*[53]. Another example pertains to soybean dregs that undergo fermentation facilitated by *Neurospora crassa*[54]. The distinctive orange hue of the fermented product arises from the generation of hyphae and spores rich in carotenoids by the fungus containing neurosporaxanthin, among other carotenoids[55].

The main goal of our study was to assess the function of purified neurosporaxanthin in mammals, using mice as a model. However, our purification strategy required large amounts of mycelium to obtain a total of 50 mg of neurosporaxanthin. As a result, we could only carry out our experiments with a total of three mice per group, resulting in a major limitation to our study. However, our promising data establishes the basis for developing methods to obtain neurosporaxanthin on a large scale. In this

respect, the culture conditions developed with carotenoid-overproducing strains of *F. fujikuroi*, which in contrast to *Neurospora* spores produce neurosporaxanthin as the major carotenoid[18], are a promising starting point. We hope that our results will stimulate further work that will allow the use of neurosporaxanthin, or fungal biomass produced in large quantities, as a functional food ingredient.

## Methods

***Fusarium fujikuroi* strain, culture conditions, and extract preparation**. The SG39 strain of *F. fujikuroi* is a carotenoid-overproducing *carS* mutant obtained by exposure of conidia of the wild-type strain IMI582289 to N-methyl-N'-nitro-N-nitrosoguanidine[56]. SG39 cultures were incubated for 5 weeks in 500 mL Erlenmeyer flasks in the dark on an orbital shaker at 150 rpm. Each flask contained 200 mL of low-nitrogen optimized medium[18], consisting of 80 g/L sucrose, 0.5 g/L of $NH_4NO_3$, and 0.5 g/L of $KH_2PO_4$, and was inoculated with $10^6$ conidia. After 5 weeks, mycelia were filtered, frozen, and vacuum dried. A sample of 10 grams of dry mycelium was extracted with acetone, as previously described[57]. The collected extract was evaporated to dryness under vacuum in a rotary evaporator at 30 °C and the pigments were dissolved in 50 mL of acetone.

**Isolation and purification of neurosporaxanthin**. We performed repetitive injections of 100 μL of the acetone extract in an HPLC system, as described, with some modifications[57]. Briefly, the chromatographic system consisted of a Waters 2695 HPLC fitted with an analytical reversed-phase Mediterranea SEA18 C18 column, 3 μm, 20×0.46 cm (Teknokroma, Barcelona, Spain), a Waters 2998 photodiode array detector, and a programmable fraction collector (Waters Fraction Collector III). The HPLC system was controlled with Empower2 software (Waters

Cromatografía, S.A., Barcelona, Spain). Carotenoid separation was achieved with a binary-gradient elution (acetone: deionized water) at a flow rate of 1 mL/min using an initial composition of 90% acetone and 10% deionized water, which was increased linearly to 100% acetone in 8 min and then maintained constant for 2 min. Initial conditions were reached in 5 min. Column and sample compartments were maintained at 25 °C and 15 °C, respectively. Detection was performed at 450 nm and online UV/Visible absorption spectra were acquired in the wavelength range of 350–600 nm. Fractions corresponding to the neurosporaxanthin peak were collected by connecting a fraction collector at the eluent outlet of the photodiode array detector. Pooled fractions were concentrated under vacuum at 30 °C and stored at −80 °C.

We monitored the purity of the neurosporaxanthin by HPLC and confirmed its identity using mass spectrometry (MS). The mass spectrum of neurosporaxanthin was obtained by HPLC with diode-array detection (DAD) coupled with atmospheric-pressure chemical ionization (APCI)-MS on a Dionex Ultimate 3000RS U-HPLC (Thermo Fisher Scientific, Waltham, MA, USA) fitted with a DAD) and linked to a micrOTOF-QII high-resolution TOF MS (UHR-TOF) with quadrupole (qQ)-TOF geometry (Bruker Daltonics, Billerica, MA, USA) equipped with an APCI source. Chromatographic conditions were the same as described above for the chromatographic characterization of neurosporaxanthin. A split post-column of 0.4 mL/min was introduced directly onto the MS ion source. The MS instrument was operated in positive ion mode, with a scan range of $m/z$ 50–1200. Mass spectra were acquired through the broadband collision-induced dissociation mode. The instrument control was performed using Bruker Daltonics Hystar 3.2. Data evaluations were performed with Bruker Daltonics DataAnalysis 4.0. The amount of purified neurosporaxanthin was spectrophotometrically calculated using a specific absorption coefficient of $1715 \, g^{-1} \, cm^{-1} \, 100 \, mL$ at 477 nm in light petroleum ether.

**Animals and diets**. All studies were performed following the guidelines published in the NIH Guide for the Care and Use of Laboratory Animals[58]. The Institutional Animal Care and Use Committee of the University of Illinois at Urbana Champaign reviewed and approved the animal protocol. Wild-type, $Bco1^{-/-}$ [13], $Bco2^{-/-}$ [12], and $Bco1^{-/-}Bco2^{-/-}$ mice were used for the experiments described. We generated congenic strains by crossbreeding $Bco1^{-/-}$ and $Bco2^{-/-}$ mice with C57BL/6 wild-type mice for 11 generations. Mice were maintained at 24 °C in a 12-h/12-h light/dark cycle with *ad libitum* access to food and water. All mice were fed a non-purified breeder diet containing 15 IU vitamin A/g diet (Teklad global 18% protein diet, Envigo, Indianapolis, IN, USA) until reaching 4 weeks of age. At 4 weeks of age, we switched the mice to a purified vitamin A-deficient, carotenoid-free standard diet (VAD Carotenoid-Free Diet) prepared by Research Diets, Inc. (New Brunswick, NJ, USA).

**Preparation of carotenoid and retinoic acid gavage solutions**. Neurosporaxanthin powder was obtained from samples purified as described above. β-cryptoxanthin was isolated and purified from beadlets that were kindly provided by OmniActive (Morristown, NJ, USA). β-carotene and *all-trans* retinoic acid powder were purchased from Thermo Fisher Scientific. The purity of the compounds was verified by HPLC. The preparation of carotenoid-containing gavage oil solutions was adapted from a method described by Deming et al.[59]. Briefly, carotenoids or retinoic acid were dissolved in suitable organic solvents, and cottonseed oil was added to the carotenoid solution to achieve a final concentration of 0.3 mg carotenoid/100 μL oil. The solvents

were evaporated in a rotary evaporator at reduced pressure and 35 °C and finished under an argon stream to prevent oxidation, as described previously[60,61]. Complete evaporation of the solvents was confirmed gravimetrically, and the solubility of the carotenoids in oil was confirmed by light microscopy using 100X magnification.

**Carotenoid and retinoic acid treatments and tissue harvesting**. Carotenoids or retinoic acid were administered to mice by gavage at a dose of 30 mg/kg body weight, or the same volume of vehicle (cottonseed oil) to control mice. For example, a 20 g mouse was gavaged a volume of 200 μL, as done previously[11,24,62–64]. Mice were euthanized 24 h after the last carotenoid gavage. At the moment of sacrifice, mice were anesthetized by an intraperitoneal injection of 80 mg ketamine and 8 mg xylazine/kg body weight, followed by blood collection directly from the heart using EDTA-coated syringes. Mice were then perfused with a saline solution (0.9% NaCl in water), after which organs were harvested, snap-frozen in liquid nitrogen, and subsequently stored at −80 °C. Blood plasma was collected by centrifugation at 2000 x $g$ for 10 min at 4 °C and immediately stored at −80 °C.

**Collection of feces**. To analyze carotenoid absorption, a subset of mice was placed in individual cages to collect feces at 12 h intervals for 3 days after a single gavage with carotenoid oil or vehicle. As previously described, grates were placed at the bottom of cages to prevent coprophagy[24,64].

**Circulating aminotransferase levels and microRNA-122 (*miR-122*) expression**. Alanine transaminase (ALT) and aspartate transaminase (AST) activities were analyzed in the plasma of mice using commercially available kits (Abcam, Cambridge, MA, USA), as per manufacturer instructions. Briefly, plasma was mixed with reaction mix and read on an automated microplate reader (Bio-Rad, Hercules, CA, USA) on kinetic mode every 3 min for 60 min at 37 °C. For analysis, two time points were chosen when all samples fell within the standard curve. Pyruvate/glutamate concentrations were calculated using the standard curve, and ALT/AST activity was determined in milliunits per mL of plasma (mU/mL).

Plasma *miR-122*, a liver-specific miRNA only found in circulation upon liver injury[26], was analyzed using RNA isolated from 70 μL of plasma with TRIzol LS reagent (Thermo Fisher Scientific). Samples were spiked with synthetic *C. elegans* miR-39 (*cel-miR-39*) (Qiagen, Hilden, Germany) as external control. We synthetized cDNA using TaqMan MicroRNA Reverse Transcription Kit (Applied Biosystems, Carlsbad, CA) and sequence-specific stem-looped primers contained in TaqMan Small RNA Assays (Thermo Fisher Scientific). Quantitative real-time PCRs were performed using TaqMan reagents, primers, and probes (Thermo Fisher Scientific). Relative *miR-122* to *cel-miR-39* expression levels were determined using the Pfaffl method considering reaction efficiencies, as previously described[65].

**HPLC analysis of carotenoids and retinoids in mouse samples**. Carotenoids and retinoids were extracted from 70 μL of plasma, or tissue homogenates in phosphate saline buffer containing 10 mg of liver or intestine under a dim yellow safety light using a method adapted for polar carotenoids. Feces, inguinal white adipose tissue (iWAT), and gonadal WAT (gWAT) were saponified prior to extraction, as described previously[66]. Plasma or tissue homogenates were mixed with 200 μL of ethanol, followed by the addition of 400 μL of acetone. Extraction of carotenoids and retinoids was performed with a mixture of hexane, ethyl acetate, and acetic acid (79.9:20:0.1 v/v). The extraction was

repeated three times, and the collected organic phases were dried using a SpeedVac vacuum concentrator (Thermo Fisher Scientific). All HPLC analyses were performed on a normal-phase Zorbax Sil (5 μm, 4.6 × 150 mm) column (Agilent Technologies, Santa Clara, CA, USA) protected with a guard column with the same stationary phase. Chromatographic separation was accomplished using an isocratic flow of 20% ethyl acetate and 0.0175% acetic acid in hexane at a flow rate of 1.0 mL/min (mobile phase). For molar quantifications of carotenoids and retinoids, the HPLC was calibrated using pure compounds. Carotenoids and retinoids were identified using standards and comparing elution times and spectra to the samples. Tissue vitamin A levels, represented as total retinoids, correspond to the sum of retinol and retinyl esters.

**mRNA isolation and quantitative PCR analysis**. Total RNA was isolated with the Direct-zol RNA MiniPrep Plus Kit (Zymo Research, Irvine, CA, USA) according to the manufacturer's instructions. A Nanodrop spectrophotometer was used to measure the concentration and purity of the RNA (Thermo Fisher Scientific). One microgram of total RNA was reverse transcribed to cDNA with the Applied BioSystems retrotranscription kit (Applied BioSystems). Quantitative real-time PCRs (RT-PCR) were performed using SYBR reagents (Applied Biosystems) and primers (Integrated DNA Technologies, Coralville, IA, US) for the following genes: intestine-specific homeobox (*Isx*, 5'-ATC TGG GCT TGT CCT TCT CC-3' and 5'-TTT TCT CTT CTT GGG GCT GA-3'), scavenger receptor class B type 1 (*Sr-b1*, 5'-TCA GAA GCT GTT CTT GGT CTG AAC-3' and 5'-GTT CAT GGG GAT CCC AGT GA-3'), BCO1 (*Bco1*, 5'-CGG AAG TAT GTG GCG GTA AA-3' and 5'-GGA GGA AAT GGA GCA GAA AA-3'), retinoic acid receptor β (*Rarβ*, 5'-CAC CAT CTC CAC TTC CTC CT-3' and 5'-GGC TCC TTC TTT TTC TTG TTC C-3'), and cytochrome P450 family 26 subfamily A member 1 (*Cyp26a1*, 5'-GGA CCT GTA CTG TGT GAG CA-3' and 5'-ATG AAG CCG TAT TTC CTG CG-3'). *β-actin* (5'-AGA GGG AAA TCG TGC GTG AC-3' and 5'-CAA TAG TGA TGA CCT GCG CGT-3') was used as a housekeeping control. Gene expression analyses were performed with the StepOnePlus RT-PCR System (Applied Biosystems) and the ΔΔCt calculation method.

**Production of recombinant BCO1 and BCO2**. The open reading frames of murine *Bco1* and *Bco2* were amplified by PCR and cloned into the respective cloning site of the pMAL-c5x expression vector (New England Biolabs, Ipswich, MA, USA) using the *XmnI* and *SbfI* restriction sites. The N-terminal of the open reading frames fused with a maltose binding protein as previously described[30]. The *E. coli* cells were grown in LB medium in the presence of 0.2% w/v glucose and 100 mg/mL ampicillin at 37 °C until the optical density at 600 nm reached 0.6–0.7. Next, cells were induced with 0.3 mM isopropyl β-D-1-thiogalactopyranoside and 30 mg/L of $FeSO_4$ for 24 h at 16 °C under vigorous shaking. Cells were harvested by centrifugation and the cell pellet was stored at –80 °C until further use.

For protein purification, the cell pellet was resuspended in column buffer (20 mM Tricine, 150 mM NaCl, 0.5 mM Tris(2-carboxyethyl)phosphine (TCEP) at pH 7.4) on ice. Then 20 μL of lysozyme (10 mg/mL), 4 μL of DNAse (Qiagen), and one protease inhibitor tablet (Roche, Basel, Switzerland) were added and incubated for 30 min on ice. Next, a French press homogenizer (Avestin emulsiflex) was used to lyse the cells. Lysates were ultracentrifuged at 125,440 x g at 4 °C for 1 h to separate soluble proteins from insoluble debris and membranes. The soluble protein extract was added to amylose resin which was previously equilibrated with 5 column volumes of column buffer. Then, the bound fraction was washed with 10 column volumes of column

buffer to remove unbound proteins from the resin. Maltose binding protein (MBP), murine BCO1, and BCO2 proteins were eluted with the elution buffer (20 mM Tricine, 150 mM NaCl, 0.5 mM TCEP, 10 mM maltose at pH 7.4). An aliquot of each fraction was separated by sodium dodecyl-sulfate polyacrylamide gel electrophoresis, and the fractions containing the protein were pooled and concentrated using centrifugal filter units (MilliporeSigma, Burlington, MA, USA). MBP was expressed and purified by the same protocol. The concentrated proteins were used in tests for enzymatic activity as described below.

**Enzyme activity assay**. We dissolved 2000 pmol of neurosporaxanthin in acetone and mixed it with reaction buffer (20 mM Tricine, 150 mM NaCl, 0.5 mM TCEP, 0.2% Triton x-100 at pH 7.4). Then 50 μg of enzyme solution was added to the mixture and incubated at 37 °C under shaking at 600 rpm in a thermomixer (Eppendorf, Hamburg, Germany) for 10 min. The reactions were stopped by the addition of 100 μL of 10% acetic acid in water (v/v), 400 μL of acetone, 400 μL of diethyl ether, and 100 μL of petroleum ether. Organic and aqueous layers were separated, collected, and evaporated in a dry vacuum centrifuge (Eppendorf). Carotenoids and retinoids were analyzed by HPLC using a Zorbax Sil column (Agilent Technologies), as outlined above.

**Statistics and reproducibility**. Data are expressed as means ± standard error of the mean (SEM). Statistical differences were analyzed using GraphPad Prism software (GraphPad Software Inc., San Diego, CA, USA). Statistical differences were evaluated with two-tailed Student *t*-testing between groups of two. To test statistical differences between more than two groups, we employed the Kruskal–Wallis *H*-test (non-parametric) applying Dunn's test correction for multiple comparisons. Statistical significance was set at $p < 0.05$.

**Reporting summary**. Further information on research design is available in the Nature Portfolio Reporting Summary linked to this article.

## Data availability
The numerical source data behind the graphs in the figures are available in the Supplementary Data file. Any remaining information can be obtained from the corresponding author upon reasonable request.

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

## Acknowledgements

We thank Molly Black and Kriti Maiya for their assistance with HPLC quantifications. We also thank Dr. Deshanie Rai (OmniActive) for providing β-cryptoxanthin for our studies. J.A.T. discloses support for the research and publication of this work from the National Institutes of Health [grant number HL147252], the United States Department of Agriculture [grant number W5002], and the Division of Nutritional Sciences-Vision 20/20 Program. D.H.M, J.A., and M.C.L. disclose support for the research and publication of this work from MCIN/AEI/10.13039/501100011033 [grant numbers 2018-101902-B-I00 and 2015-69613-R] and from Junta de Andalucía [grant numbers P10-CTS-6638 and P20-01243]. D.H.M, J.A., and M.C.L. are members of the Spanish Carotenoid Network (CaRed), grant RED2022-134577-T, funded by MCIN/AEI/10.13039/501100011033. A.P.M. is a recipient of the NIH Ruth L. Kirschstein National Research Service Award Fellowship [grant number AT012145]. O.P.R. was a recipient of a MCI/AEI fellowship.

## Author contributions

A.P.M., J. Avalos, J. Amengual, M.C.L., and J.V.L. conceptualized the experiments. A.P.M., D.H.M., S.B., and O.P.R. performed the experiments. A.P.M. drafted the manuscript. A.P.M., J. Avalos, J. Amengual, J.V.L., D.H.M., and M.C.L. edited and approved the manuscript.

## Competing interests

The authors declare no competing interests.
