## [Peer Review File · Communications Biology]

Reviewers' comments:

Reviewer #1 (Remarks to the Author):

This is an original research paper describing bioavailability and potential as a provitamin A of neurosporaxanthin, a xanthophyll produced in fungi, in mice. While research methods are sound and conclusions are logical based on presented data, it is difficult to make firm conclusions based on only 3 mice per group in some cases.

This reviewer finds following concerns.

1. Data presented here are generated in mice. Therefore, use of "mammal" in the title is overly broad.
2. What are the total retinoids reported in Fig 3? How long after the last gavage of carotenoids were the mice euthanized? What is the interpretation of increased retinoids in intestine after the repeated carotenoids gavage? Is it a temporary increase due to increased retinoid formation from carotenoids or more stable increase due to increased retinoid levels in intestine?
3. Lines 294-296. While neurosporaxanthin did not increase liver enzymes after 10 days of treatment, it is premature to conclude that "neurosporaxanthin does not cause hepatic toxicity" as experiment is not designed to study this question. For such conclusion, longer term and variable doses need to be used. However, if this is the dose that a human would be taking, it would be reasonable to make an interpretation that is narrower to address this question.
4. Lines 330 – 342. It would have been better to test inability of neurosporaxanthin to stimulate RARs directly by binding studies or by using cells transfected with RARs. It is more difficult to test this in mice due to many variables as authors acknowledge. It is unclear why the comparison is between KO mice given neurosporaxanthin given for 10 days and wt mice given RA for one time. It would make more sense to compare KO mice vs. WT mice given neurosporaxanthin since WT mice would convert neurosporaxanthin to RA?
5. ANOVA assumes equal variance and normal distribution of data. With only three data points per group, non-parametric test may be more appropriate. Please indicate what the error bars denote - SEM or SD?

Reviewer #2 (Remarks to the Author):

Review

The authors present an analysis of the uptake and metabolism of the fungal carotenoid neurosporaxanthin in mice and explore the utility of this carotenoid as a potential food additive. They find that neurosporaxanthin accumulates at higher levels than other carotenoids (e.g. Beta-carotene) in the livers and plasma of BCO1 and BCO2 KO mice and suggest that this increased bioavailability results from the greater polarity/hydrophilicity of this carotenoid. The authors go on to show that both BCO1 and BCO2 can cleave neurosporaxanthin in vitro and that this compound can function as a provitamin A carotenoid in vivo. Finally, they present evidence that intact neurosporaxanthin is not a retinoic acid receptor agonist and that neurosporaxanthin accumulation is not hepatotoxic. Taken together their results indicate that neurosporaxanthin could be added to mammalian diets as a safe and effective vitamin A source.

The manuscript was a pleasure to read. The experiments are well conceived, executed, and communicated. I have only one major concern/suggestion:

Neurosporaxanthin uptake and elimination were tested by gavage with a cottonseed oil vehicle. I am curious if and how the polarity or complexity of the vehicle might influence the relative uptake of various carotenoid types. Is the finding of high relative uptake likely to hold if neurosporaxanthin and other carotenoids are integrated into a food mixture? I don't think additional experiments are necessary, but I think it would be helpful to address, in the discussion, the potential effects of delivery

vehicle and briefly review what has been observed for other types of carotenoids of varying polarity.

Minor comments:

Lines 90-94: In this paragraph, please also set up the reasoning for including the transaminase and gene expression assays.

Line 144: Please explain what the limitations of the purified neurosporaxanthin are. Do you simply mean the amount of material available?

Lines 159-161: Why was this dose selected? Is this comparable to the levels that might be encountered in food or a supplement?

Lines 171-178: Reading the paper in order, the purpose of these assays was not immediately clear. As noted above, please add this information to the last paragraph of the introduction.

Line 260-261: 500 hundred injections, Wow!

Line 290: Why 10 consecutive days?

Lines 305-307: Was this result unexpected? What is the Vitamin A content of the VAD carotenoid-free diet?

Line 337-342: Why is the positive control in WT mice rather than the double KO. These experiments are still useful, but it is a bit strange. Are gene expression levels in the double KO and WT vehicle-treated the same?

Line 396: "Only a handful of carotenoids possess provitamin A activity". I would suggest noting that this is in mammals. The picture may be more complex for vertebrates, but this question is not well resolved in other groups.

Line 404: Any evidence of the high antioxidant capacity of neurosporaxanthin in the current study?

Line 408: It would be helpful to the reader to add a few more words of description of oncom. I had to google it!

Reviewer #3 (Remarks to the Author):

The manuscript from Miller et al. provides evidence that the fungal carboxylic carotenoid neurosporaxanthin is absorbed and metabolized by mammals, specifically by mice. The authors also provide both in vitro and in vivo evidence that neurosporaxanthin can be cleaved by mammalian BCO1 and by BCO2 to form vitamin A.

The manuscript is well written and interesting. The experimental work is very rigorous and technically sound. The data that are reported convincingly support the conclusions stated above. I have no reservations regarding the quality of the work or with the above conclusions. This manuscript will undoubtedly be of interest to investigators working to understand the metabolism and actions of carotenoids.

I do however have reservations regarding the Discussion section and its contents. This all centers on the significance of the work and the authors' apparent attempt to hype the importance of the new information they have obtained. The authors' data rigorously establishes that neurosporaxanthin is a provitamin A carotenoid; one that is especially well absorbed by the mouse gut. These findings need

to be reported in the literature. However, is there broader significance associated with these findings? The authors mention in the first paragraph of the Discussion that a fungal species closely related to the one that is being used as a source of neurosporaxanthin in the present study is being tried as source of protein for human consumption. The authors imply that their new findings may be of importance for understanding health benefits should this fungus ultimately be incorporated into human diets. But this first paragraph and the remainder of the discussion is so loosely and vaguely written and so poorly referenced, one might critically take this to be nothing more than casual speculation. This idea is simply not well developed in the text and this detracts from the manuscript. The next 4 paragraphs of the discussion consider the authors' new findings and their true scientific significance. This discussion is appropriated for the new data being provided. However, the final 3 paragraphs provide a lengthy discourse regarding how neurosporaxanthin synthesizing fungi may be identified, propagated and used in the creation/processing of human food. These final 3 paragraphs are simply speculation that does not flow directly from the authors' new findings. Possibly these issues are of interest to the authors but they are well outside the focus of the new data being reported. Possibly, this is a detailed plan for a next step of this project but these paragraphs do not belong in the present manuscript. The Discussion needs to be heavily edited and greatly shortened to be consistent with the new information being reported.

Referee expertise:

Referee #1: retinoid and carotenoid metabolism

Referee #2: carotenoid metabolism

Referee #3: retinoid and carotenoid metabolism

Reviewers' comments:

Reviewer #1 (Remarks to the Author):

This is an original research paper describing bioavailability and potential as a provitamin A of neurosporaxanthin, a xanthophyll produced in fungi, in mice. While research methods are sound and conclusions are logical based on presented data, it is difficult to make firm conclusions based on only 3 mice per group in some cases.

Answer: We thank the reviewer for the positive evaluation of the manuscript. We agree with the difficulty of establishing firm conclusions based on only 3 mice per group as the amount of pure neurosporaxanthin was the limiting factor for increasing the sample size in our studies. We now acknowledge this caveat in the discussion section of the revised version of the manuscript (Lines: 445-447).

This reviewer finds following concerns.

1. Data presented here are generated in mice. Therefore, use of “mammal” in the title is overly broad.

Answer: We agree with the reviewer that the use of “mammal” is overly general. In its place, we now use the word “mice” in the title.

2. What are the total retinoids reported in Fig 3?

Answer: We appreciate the reviewers comment pertaining to the clarity of retinoid determinations and their interpretations. The use of “total retinoids” in Figure 3 corresponds to the sum of retinol and retinyl esters as determined by HPLC. We have clarified this in the methodology section by adding “Tissue vitamin A levels, represented as total retinoids, corresponds to the sum of retinol and retinyl esters.” (Lines: 220-221).

How long after the last gavage of carotenoids were the mice euthanized?

Answer: We apologize for omitting this information. In the current version of the manuscript, we include the statement in the methodology “Mice were euthanized 24 h after the last carotenoid gavage.” (Lines: 177-178).

What is the interpretation of increased retinoids in intestine after the repeated carotenoids gavage? Is it a temporary increase due to increased retinoid formation from carotenoids or more stable increase due to increased retinoid levels in intestine?

Answer: The intestine is the first organ in contact with dietary carotenoids. Therefore, we assume that an increase in intestinal retinoid levels is a temporary increase due to retinoid formation in the enterocyte, where BCO1 is highly expressed. Since the intestine is not a primary storage organ for

retinoids, these increased concentrations are likely transient, whereas in tissues like the liver or adipose tissue, retinoids are stored.

In the previous version of our manuscript, however, we observed a trend towards increasing hepatic retinoid levels. In our current version, we measured retinoids in two different depots of white adipose tissue (WAT) (Figure 3E). Results in WAT strengthen our previous findings in the intestine and suggest that neurosporaxanthin has a comparable capacity to β -carotene to serve as vitamin A precursor.

3. Lines 294-296. While neurosporaxanthin did not increase liver enzymes after 10 days of treatment, it is premature to conclude that “neurosporaxanthin does not cause hepatic toxicity” as experiment is not designed to study this question. For such conclusion, longer term and variable doses need to be used. However, if this is the dose that a human would be taking, it would be reasonable to make an interpretation that is narrower to address this question.

Answer: We agree with the reviewer that we somewhat overstated that neurosporaxanthin does not cause hepatic toxicity. While we do not have the means to conduct a longer-term experiment with variable doses, we have added more data to the manuscript to illustrate that the mice gavaged for 10 days with neurosporaxanthin did not present changes in various parameters commonly associated with toxicity in mice. Additionally, we have now included the expression levels of miR122, a relatively new biomarker of liver toxicity. We have also included morphometric and food intake data that support our assumption that the administration of neurosporaxanthin, at least under our experimental conditions, does not cause apparent signs of toxicity. These data are now presented in the new Table 1 (Line: 457). Regardless, we have re-worded our results regarding toxicity to avoid misinterpretation of our data (Lines: 321-323).

4. Lines 330 – 342. It would have been better to test inability of neurosporaxanthin to stimulate RARs directly by binding studies or by using cells transfected with RARs. It is more difficult to test this in mice due to many variables as authors acknowledge.

Answer: The reviewer raises a valid point. We agree that binding studies would be the gold standard to unequivocally establish whether neurosporaxanthin activates RARs. However, our in vivo data discouraged us to pursue this route. More specifically, we did not observe changes on classical RAR target genes despite the accumulation of high amounts of neurosporaxanthin in the liver of *Bco1^{-/-}Bco2^{-/-}* mice fed 10 days. Together, these findings suggest that neurosporaxanthin does not activate RAR signaling. This assumption is in line with work from the Harrison’s lab that showed that long chain beta-apocarotenoids do not activate RARs (PMID: 20404052).

In the revised version of the manuscript, we reworded the result section heading and text to diminish the enthusiasm of our original version.

It is unclear why the comparison is between KO mice given neurosporaxanthin given for 10 days and wt mice given RA for one time. It would make more sense to compare KO mice vs. WT mice given neurosporaxanthin since WT mice would convert neurosporaxanthin to RA?

Answer: We apologize for the lack of clarity in the presentation of these results. In agreement with the Reviewer 1 and 2, we have now separated the results on carotenoid and retinoid accumulation (current Figure 3) from those results on gene expression (current Figure 5). We have also removed the results of mice treated with retinoic acid.

5. ANOVA assumes equal variance and normal distribution of data. With only three data points per group, non-parametric test may be more appropriate. Please indicate what the error bars denote - SEM or SD?

Answer: We appreciate the reviewer's comment and have now included the text "Values are means \pm SEMs" in figure legends, when necessary.

As the Reviewer pointed out, we cannot assume normality, and therefore, we have re-analyzed our results using Kruskal-Wallis with Dunn's post-hoc analysis. As a result, some of our results lost significance.

Additionally, we have edited the graphs to display the individual values for each mouse and include the p values between groups, for transparency.

We have also re-worded the statistical analysis section (Lines: 277-279) and edited the results and figure legends, accordingly.

Reviewer #2 (Remarks to the Author):

The authors present an analysis of the uptake and metabolism of the fungal carotenoid neurosporaxanthin in mice and explore the utility of this carotenoid as a potential food additive. They find that neurosporaxanthin accumulates at higher levels than other carotenoids (e.g. Beta-carotene) in the livers and plasma of BCO1 and BCO2 KO mice and suggest that this increased bioavailability results from the greater polarity/hydrophilicity of this carotenoid. The authors go on to show that both BCO1 and BCO2 can cleave neurosporaxanthin in vitro and that this compound can function as a pro-vitamin A carotenoid in vivo. Finally, they present evidence that intact neurosporaxanthin is not a retinoic acid receptor agonist and that neurosporaxanthin accumulation is not hepatotoxic. Taken together their results indicate that neurosporaxanthin could be added to mammalian diets as a safe and effective vitamin A source.

The manuscript was a pleasure to read. The experiments are well conceived, executed, and communicated.

Answer: We appreciate the positive evaluation of our manuscript.

I have only one major concern/suggestion:

Neurosporaxanthin uptake and elimination were tested by gavage with a cottonseed oil vehicle. I am curious if and how the polarity or complexity of the vehicle might influence the relative uptake of various carotenoid types. Is the finding of high relative uptake likely to hold if neurosporaxanthin and other carotenoids are integrated into a food mixture? I don't think additional experiments are necessary, but I think it would be helpful to address, in the discussion, the potential effects of delivery vehicle and briefly review what has been observed for other types of carotenoids of varying polarity.

Answer: We agree that the polarity of carotenoids (β -carotene, β -cryptoxanthin, and neurosporaxanthin) could affect their relative uptake. This was a major concern for us during the conceptualization of the experiments. Hence, we ensured that the three compounds were completely soluble in the cottonseed oil by gravimetric analysis and by examining the oil under 100X magnification. We did not observe any residues in the any of the preparations employed in our study.

Now, we address the rationale behind the use of cottonseed oil in the current version of the manuscript, where we introduce a brief overview covering the influence of fat type on carotenoid absorption and our choice of cottonseed oil as a vehicle (Lines :391-396).

Additionally, we provided some more clarifications about the solubility of carotenoids in our preparations in the methods section (Lines: 166-168 and 172-173).

Minor comments:

Lines 90-94: In this paragraph, please also set up the reasoning for including the transaminase and gene expression assays.

Answer: We agree with the reviewer that the evaluation of the toxicity of neurosporaxanthin was not prefaced. It is now stated in the last paragraph of the introduction that it was one of our endpoints. (Lines: 104-106).

Additionally, in the current version of the manuscript we evaluate the expression of miR-122 in plasma, which is considered a highly sensitive marker of liver damage (PMID: 19246379).

Line 144: Please explain what the limitations of the purified neurosporaxanthin are. Do you simply mean the amount of material available?

Answer: We apologize for the lack of clarity. Indeed, we could only purify a total of 50 mg of neurosporaxanthin. In the current version of the manuscript, we noted this limitation in the discussion section (Lines: 443-446).

Lines 159-161: Why was this dose selected? Is this comparable to the levels that might be encountered in food or a supplement?

Answer: Over the years, we have consistently utilized the dose of 30 mg/kg of body weight to gavage a variety of retinoids and carotenoids (PMIDs: 22637576, 24852372, 34742949, 32963037, 24106281). We have added these citations in the methodology to provide a rationale for the dosage.

30 mg/kg of body weight is a high dose, and therefore, it is unlikely it could be found in a food or a supplement. However, it allows us to reliably detect carotenoids/retinoids with our HPLC system, allowing us to perform accumulation and biodistribution studies.

Lines 171-178: Reading the paper in order, the purpose of these assays was not immediately clear. As noted above, please add this information to the last paragraph of the introduction.

Answer: Thanks for your suggestions, we have clarified this aspect in the last paragraph of the introduction (Lines 104-106).

Line 260-261: 500 hundred injections, Wow!

Answer: Neurosporaxanthin is not a commercially available carotenoid. Therefore, a particularly challenging aspect was to obtain enough neurosporaxanthin to carry out these studies.

Line 290: Why 10 consecutive days?

Answer: We chose ten days of intervention based on prior experiments utilizing the synthetic retinoid fenretinide, which inhibits the formation of vitamin A from β -carotene. We performed daily gavages with 30 mg/kg of fenretinide and observed that the partial inhibition of BCO1 in mice led to differences in vitamin A stores in tissues (PMID: 34742949). Hence, we decided to adapt these experimental setting to this study (time and dosage).

Now we include data on vitamin A stores in two different adipose tissue depots showing that, indeed, 10 days were enough to alter vitamin A levels in this organ.

Lines 305-307: Was this result unexpected?

Answer: Indeed, the lack of change in hepatic vitamin A stores was surprising because we surmised that the administration of either β -carotene or neurosporaxanthin would increase hepatic vitamin A levels compared to mice that received vehicle.

These results indicate that even though the mice were only four weeks old, they had stored large amounts of vitamin A in the liver. In the new version of the manuscript, we have now analyzed vitamin A stores in the adipose tissue, which in combination with the vitamin A levels in the intestine, unequivocally show that β -carotene and neurosporaxanthin produce vitamin A in wild-type mice.

What is the Vitamin A content of the VAD carotenoid-free diet?

Answer: We utilize customized diets prepared by research diets. They utilize a vitamin A deficient mixture, and therefore, there is no vitamin A in the diet. When we order new batches, we consistently validate them with our HPLC system.

Line 337-342: Why is the positive control in WT mice rather than the double KO. These experiments are still useful, but it is a bit strange. Are gene expression levels in the double KO and WT vehicle-treated the same?

Answer: Thanks to the feedback by Reviewer 1 and 2, we decided to remove the retinoic acid data as it could confuse the readers. In the current version of the manuscript, we have replaced the RA experiment for WT fed neurosporaxanthin and separated our results on carotenoid/retinoid accumulation in tissues (current Figure 3) from gene expression results (current Figure 5).

Line 396: "Only a handful of carotenoids possess provitamin A activity". I would suggest noting that this is in mammals. The picture may be more complex for vertebrates, but this question is not well resolved in other groups.

Answer: Thank you for the suggestion. We now have edited this statement, as suggested (Line: 424).

Line 404: Any evidence of the high antioxidant capacity of neurosporaxanthin in the current study?

Answer: We have reported antioxidant effects of neurosporaxanthin in cell culture conditions (Line 433), however, we have not explored the antioxidant effects in our mouse experiments, as these assays were outside of the scope of our present study.

Line 408: It would be helpful to the reader to add a few more words of description of oncom. I had to google it!

Answer: We agree with the reviewer. In the current version, we added further context about oncom and included a recent publication demonstrating the presence of neurosporaxanthin in these fermented products (Lines: 434-442).

Reviewer #3 (Remarks to the Author):

The manuscript from Miller et al. provides evidence that the fungal carboxylic carotenoid neurosporaxanthin is absorbed and metabolized by mammals, specifically by mice. The authors also provide both in vitro and in vivo evidence that neurosporaxanthin can be cleaved by mammalian BCO1 and by BCO2 to form vitamin A.

The manuscript is well written and interesting. The experimental work is very rigorous and technically sound. The data that are reported convincingly support the conclusions stated above. I have no reservations regarding the quality of the work or with the above conclusions. This manuscript will undoubtedly be of interest to investigators working to understand the metabolism and actions of carotenoids.

I do however have reservations regarding the Discussion section and its contents. This all centers on the significance of the work and the authors' apparent attempt to hype the importance of the new information they have obtained. The authors' data rigorously establishes that neurosporaxanthin is a provitamin A carotenoid; one that is especially well absorbed by the mouse gut. These findings need to be reported in the literature. However, is there broader significance associated with these findings? The authors mention in the first paragraph of the Discussion that a fungal species closely related to the one that is being used as a source of neurosporaxanthin in the present study is being tried as source of protein for human consumption. The authors imply that their new findings may be of importance for understanding health benefits should this fungus ultimately be incorporated into human diets. But this first paragraph and the remainder of the discussion is so loosely and vaguely written and so poorly referenced, one might critically take this to be nothing more than casual speculation. This idea is simply not well developed in the text and this detracts from the manuscript. The next 4 paragraphs of the discussion consider the authors' new findings and their true scientific significance. This discussion is appropriated for the new data being provided. However, the final 3 paragraphs provide a lengthy discourse regarding how neurosporaxanthin synthesizing fungi may be identified, propagated and used in the creation/processing of human food. These final 3 paragraphs are simply speculation that does not flow directly from the authors' new findings. Possibly these issues are of interest to the authors but they are well outside the focus of the new data being reported. Possibly, this is a detailed plan for a next step of this project but these paragraphs do not belong in the present manuscript. The Discussion needs to be heavily edited and greatly shortened to be consistent with the new information being reported.

Answer: We agree with the reviewer's assessment of the discussion. In the current version of the manuscript, we remove the speculative sections of the discussion to focus on the findings of the study. This part of the discussion has refocused on the biotechnological possibilities of neurosporaxanthin and its lack of toxicity, further supported by the presence of this carotenoid in foods derived from *Neurospora* fermented soybean products consumed in Asia, and on the existence of already developed conditions of production in high concentrations with appropriate *Fusarium* strains. Regarding the presence of neurosporaxanthin in oncom, a paper has just been published in which this is demonstrated by HPLC analyses, and the reference has been added in the discussion. We believe that this part of the discussion dedicated to the biotechnological implications of our study is appropriate in accordance with the results obtained, and that it has now been more adequately focused.

REVIEWERS' COMMENTS:

Reviewer #1 (Remarks to the Author):

Revised manuscript fully addresses concerns raised by this reviewer. The additional adipose tissue data provide stronger evidence that neurosporaxanthin is a provitamin A carotenoid in mice.

Reviewer #2 (Remarks to the Author):

The authors have addressed my concerns from the previous review. The methodology and results are now clear, and the conclusions and discussion have been revised in a way that acknowledges the limitations of the study.

There is one comment and a few minor typos that should be addressed:

Figure 2 panel F – This pane is titled “Hepatic Carotenoid Excretion”. I believe this more accurately is “Hepatic Carotenoid Accumulation”

Line 240 – This line refers to the BCO1 and BCO2 transcripts there for the gene symbols should be written in title case and italicized.

Lines 391-396 – Here it should be acknowledged that delivering the carotenoid dissolved in oil is likely to be among the most favorable conditions for uptake and absorption. Therefore, these estimates of uptake are likely on the high end of what could be expected from food products containing neurosporaxanthin.

Line 446 – “three mice group” - Three mice per group?

Line 537-530 – part of the identifier is missing - Junko Yabuzaki, Carotenoids Database: structures, chemical fingerprints and distribution among organisms, Database (Oxford), 2017 (2017) bax004.

Line 675-676 – This is an issue for the copy editor, but through the journal, names are written in different ways. Here it is abbreviated with periods, in other places it is abbreviated without the period, and still others the full names are presented.

Reviewer #3 (Remarks to the Author):

The authors have addressed well and fully my earlier concern.

REVIEWERS' COMMENTS:

Reviewer #1 (Remarks to the Author):

Revised manuscript fully addresses concerns raised by this reviewer. The additional adipose tissue data provide stronger evidence that neurosporaxanthin is a provitamin A carotenoid in mice.

Answer: We appreciate your help at improving our manuscript.

Reviewer #2 (Remarks to the Author):

The authors have addressed my concerns from the previous review. The methodology and results are now clear, and the conclusions and discussion have been revised in a way that acknowledges the limitations of the study.

There is one comment and a few minor typos that should be addressed:

Answer: We appreciate your help at improving our manuscript. In the current version of the Manuscript, we have addressed these concerns.

Figure 2 panel F – This pane is titled “Hepatic Carotenoid Excretion”. I believe this more accurately is “Hepatic Carotenoid Accumulation”

Answer: We edited the figure title, accordingly.

Line 240 – This line refers to the BCO1 and BCO2 transcripts there for the gene symbols should be written in title case and italicized.

Answer: We updated the format, as suggested.

Lines 391-396 – Here it should be acknowledged that delivering the carotenoid dissolved in oil is likely to be among the most favorable conditions for uptake and absorption. Therefore, these estimates of uptake are likely on the high end of what could be expected from food products containing neurosporaxanthin.

Answer: We have acknowledged this issue by adding the following sentence: “Regardless, we acknowledge that delivering purified carotenoids in oils probably results in greater uptake rates than when carotenoids are present in foods matrices.” (Lines: 223-226)

Line 446 – “three mice group” - Three mice per group?

Answer: Done.

Line 537-530 – part of the identifier is missing - Junko Yabuzaki, Carotenoids Database: structures, chemical fingerprints and distribution among organisms, Database (Oxford), 2017 (2017) bax004.

Answer: We re-formatted the bibliography on Endnote. This issue was automatically resolved.

Line 675-676 – This is an issue for the copy editor, but through the journal, names are written in different ways. Here it is abbreviated with periods, in other places it is abbreviated without the period, and still others the full names are presented.

Answer: Please see comment above.

Reviewer #3 (Remarks to the Author):

The authors have addressed well and fully my earlier concern.

Answer: Thanks for your valuable feedback.